# Geriatric Conditions and Functional Disability among a National Community-Dwelling Sample of Older Adults in India in 2017–2018

**DOI:** 10.3390/geriatrics6030071

**Published:** 2021-07-21

**Authors:** Supa Pengpid, Karl Peltzer

**Affiliations:** 1ASEAN Institute for Health Development, Mahidol University, Salaya, Phutthamonthon, Nakhon Pathom 73170, Thailand; supa.pen@mahidol.ac.th or; 2Department of Research Administration and Development, University of Limpopo, Sovenga 0757, South Africa; 3Department of Psychology, University of the Free State, Bloemfontein 9300, South Africa; 4Department of Psychology, College of Medical and Health Science, Asia University, Taichung 41354, Taiwan

**Keywords:** geriatric conditions, disability, chronic conditions, older adults, India

## Abstract

This study aimed to determine the prevalence of geriatric conditions and their association with disability in older community-dwelling adults in India. The cross-sectional sample consisted of 31,477 individuals (≥60 years) from the Longitudinal Ageing Study in India (LASI) Wave 1 in 2017–2018. Geriatric conditions assessed included injurious falls, impaired cognition, underweight, dizziness, incontinence, impaired vision and impaired hearing. More than two in five participants (44.3%) had no geriatric condition, 32.7% had one, 15.9% two and 7.1% had three or more geriatric conditions; 26.9% were underweight, 14.5% dizziness, 13.7% had impaired vision, 9.6% impaired hearing, 9.3% impaired cognition, 8.2% major depressive disorder, 5.7% injurious falls, 4.0% incontinence, and 7.4% had Activity of Daily Living (ADL) dependencies. In logistic regression analysis, adjusted by sociodemographic factors and the number of chronic conditions, we found a higher number of geriatric conditions, and a higher number of chronic conditions were associated with ADL dependencies. In a model adjusted for sociodemographic factors and the type of chronic conditions, we found that a higher number of geriatric conditions and heart disease, stroke, and bone or joint disorder were positively associated with ADL dependencies. The odds of ADL dependencies increased with impaired cognition, impaired vision, impaired hearing, and major depressive disorder. Impaired cognition, incontinence, impaired vision and major depressive disorder were positively associated with dressing, bathing, eating, transferring, and toileting dependency. In addition, impaired hearing was associated with transferring and toileting dependency. More than half of older adults in India had at least one geriatric condition. The prevalence of geriatric conditions was as high as the prevalence of chronic conditions, which in some cases were associated with disability. Geriatric conditions should be included in health care management.

## 1. Introduction

“Health status in aging is a result of many factors, including chronic diseases of aging and many other prevalent geriatric conditions that cannot be defined as classic diseases because they do not manifest in a single organ system or result from a single pathologic cause” [1]. Geriatric conditions may include injurious falls, cognitive impairment, delirium, incontinence, hearing and visual impairment, and frailty [1], and may be better predictors of mortality than the number or presence of disease entities, but are frequently not taken care of in health care services or included in epidemiological research [2].

In a study among older adults (≥65 years) in USA, the prevalence of having one or more of seven geriatric conditions (cognitive impairment 7.3%, injurious falls 9.6%, incontinence 12.7%, underweight 2.9%, dizziness 13.4%, vision impairment 8.0% and hearing impairment 25.7%) was 49.9% [3]. “India is projected to become the world’s most populous nation by 2028, with a population of some 1.45 billion” [4]. The share of “population over the age of 60 is projected to increase from 8 percent in 2015 to 19 percent in 2050. By the end of the century, the elderly will constitute nearly 34 percent of the total population in the country” [5]. “India has 112 million elderly people with multiple physical, social, psychological, and economic problems with unmet needs in all domains of health, e.g., 40 million suffer from poor vision, 3.7 million suffer dementia, 1.6 million annual stroke cases, 1 in 3 suffer from arthritis, 1 in 3 has hypertension, 1 in 4 suffer from depression, 1 in 5 has diabetes, 1 in 5 has auditory problems, 1 in 10 falls and sustains a fracture” [6]. In a study among older adults (N = 407) in India, 83.3% had visual impairment, 14.5% urinary complaints, 63.1% self-reported hearing impairment, 5.2% suffered falls in the past 6 months, and 20.4% were underweight [7].

Having geriatric conditions has been found to be associated with Activities of Daily Living (ADL) dependencies [3], and having chronic conditions and geriatric conditions can further cause disability [8]. Considering differences in socioeconomic contexts, culture, retirement and leisure in low resourced countries, such as in India, an understanding of geriatric conditions and disability among older adults in India is important. The “National Policy on Older Persons” (NPOP) in India has been instituted for improving quality of life of elderly in India [5]. One means to be used for evaluating the quality of life of older adults in India is by assessing and monitoring geriatric conditions. To our knowledge, we could not find any study on geriatric conditions among older adults in India, which prompted this study. This study aimed to determine the prevalence of geriatric conditions, and their association with disability in older community-dwelling adults in India.

## 2. Method

### Sample and Procedures

This secondary data analysis utilised data from the cross-sectional and nationally representative “Longitudinal Ageing Study in India (LASI) Wave 1, 2017–2018”; “the overall household response rate is 96%, and the overall individual response rate is 87%.” [9] In a household survey, “interview, physical measurement and biomarker data were collected from individuals aged 45 and above and their spouses, regardless of age” [9], but we restricted out sample to persons 65 years and older with complete geriatric condition assessment. The study was approved by the “Indian Council of Medical Research (ICMR) Ethics Committee and written informed consent was obtained from the participants” [9]. All methods were carried out in accordance with relevant guidelines and regulations.

## 3. Measures

### 3.1. Outcome Variable

Activities of Daily Living (ADL) (six items) included difficulties with dressing, ambulating, bathing, eating, transferring, and toileting (Yes/No) [10]; Cronbach alpha was 0.86 for the ADL scale in this study. The ADL instrument has shown acceptable validity for the geriatric population in India [11]. “Our definition of ADL dependency required respondents to both have difficulty with and receive assistance for the task. ‘Difficulty’ included the inability to perform the task because of a health or memory problem” [3].

### 3.2. Exposure Variables

Geriatric conditions included: (1) injurious falls in the past 2 years; (2) impaired cognition; (3) underweight (measured body mass index (BMI) < 18.5 kg/m^2^); (4) dizziness (persistent or troublesome dizziness or light headedness in past two years); (5) incontinence (diagnosed by care provider); (6) impaired vision (poor or very poor far and near eyesight despite use of corrective lenses); (7) hearing impairment (diagnosed with any hearing or ear-related problem or condition); and (8) major depressive disorder. Impaired cognition was defined as 10th percentile of the total score on tests involving orientation, immediate and delayed word recall, serial 7 s, and backward counting (0–32) [9,12]. Major depressive disorder in the past 12 months was assessed with the Health and Retirement Study (HRS) Composite International Diagnostic Interview short form (CIDI-SF) [13], using criteria of the Diagnostic and Statistical Manual of Mental Disorders (DSM-5) [14]. Study respondents were required to “endorse either anhedonia or depressed mood for most of the day for most of a 2-week period or more”, and those who fulfilled this criterion “completed an additional seven symptoms: lost interest, feeling tired, change in weight, trouble with sleep, trouble concentrating, feeling down, and thoughts of death” [15]. “Those with a score ≥ 3 were considered to meet the criteria for having MDD in the previous 12 months; MDD symptomology scores ranged from 0 to 7” [15].

### 3.3. Covariates

Sociodemographic variables consisted of level of education (none, ≥1 years), age group, sex (male, female), residential status, religion, and marital status. Subjective socioeconomic status was assessed with the question, “Please imagine a ten-step ladder, where at the bottom are the people who are the worst off—who have the least money, least education, and the worst jobs or no jobs, and at the top of the ladder are the people who are the best off—those who have the most money, most education, and best jobs. Please indicate the number (1–10) on the rung on the ladder where you would place yourself” [9]. Steps 1–3 on the socioeconomic ladder were defined as low, 4–5 as medium, and 6–10 as high socioeconomic status.

Chronic conditions were sourced from the questions, “Has any health professional ever told you that you have…?”: (1) “chronic lung disease such as asthma, chronic obstructive pulmonary disease/chronic bronchitis or other chronic lung problems; (2) cancer or malignant tumor; (3) chronic heart diseases such as coronary heart disease (heart attack or myocardial infarction), congestive heart failure, or other chronic heart problems; (4) diabetes or high blood sugar; (5) bone or joint disorder (arthritis or rheumatism, Osteoporosis or other bone/joint diseases); and (6) stroke.” (Yes, No) [9]. Angina was assessed with the “World Health Organization’s Rose angina questionnaire” [16], defined on the basis of “discomfort at walking uphill or hurrying, or at an ordinary pace on level ground. Furthermore, the pain should be located at the sternum or in the left chest and arm, causing the patient to stop or slow down, and the pain should resolve within 10 min when the patient stops or slows down” [17].

## 4. Data Analysis

Descriptive statistics were applied to describe sociodemographic information, geriatric and chronic conditions. Pearson chi-squared tested for differences in proportions. Logistic regressions were used to estimate the associations between geriatric conditions and ADL dependencies. *p* < 0.05 was accepted as significant, missing values were excluded, and no multi-collinearity was found. Statistical analyses were conducted using “STATA software version 15.0 (Stata Corporation, College Station, TX, USA)”, taking the complex study design into account.

## 5. Results

### 5.1. Sample Characteristics

The sample included 31,477 older adults (60 years and older, median 68 years); 52.5% were female and 47.5% male. The majority (70.5%) of the participants were rural dwellers, 56.5% had no schooling, 61.6% were married, 82.2% were Hindus, and 39.1% had low subjective socioeconomic status. One in five participants (19.7%) had a bone or joint disorder, 14.3% diabetes, 8.5% chronic lung disease, 5.2% heart disease, and 7.4% had at least one ADL dependency. Two in five participants (43.6%) had one or more chronic conditions; 32.7% had one, 15.9% two, and 7.1% had three or more geriatric conditions. The number of geriatric conditions differed by six sociodemographic indicators but not religion, by five of seven chronic conditions and ADL dependencies (see Table 1).

The distribution of geriatric conditions by age group are shown in Table 2. The highest prevalence was found for underweight (27.2%), followed by impaired vision (13.7%), dizziness (12.7%), impaired cognition (10.7%), and impaired hearing (9.6%). The prevalence of underweight and incontinence was significantly higher among men than women (*p* < 0.001), and the prevalence of impaired cognition, dizziness, impaired vision and major depressive disorder was significantly higher in women than men (*p* < 0.001) (see Table 2).

### 5.2. Associations with ADL Dependencies

In logistic regression analysis, adjusted by sociodemographic factors and the number of chronic conditions, we found a higher number of geriatric conditions, and a higher number of chronic conditions were associated with ADL dependencies. In a model adjusted for sociodemographic factors and the type of chronic conditions, we found that a higher number of geriatric conditions and heart disease, stroke, and bone or joint disorder were positively associated with ADL dependencies, in the overall and gender stratified models (see Table 3).

### 5.3. Associations between Geriatric Conditions and ADL Dependencies

The odds of ADL dependencies increased with impaired cognition, impaired vision, impaired hearing, and major depressive disorder in the overall and male sex model, while in the female sex model ADL dependencies increased with impaired cognition, impaired vision, and impaired hearing. Impaired cognition, incontinence, impaired vision and major depressive disorder were positively associated with dressing, bathing, eating, transferring, and toileting dependency. In addition, impaired hearing was associated with transferring and toileting dependency.

In gender stratified analysis, incontinence and major depressive disorder were among men but not among women associated with ADL dependencies. Incontinence was among men but not among women associated with dressing, bathing, eating, transferring and toileting dependency. Major depressive disorder was among both men and women associated with dressing dependency, among men but not women with eating and toileting dependency and among women but not men with transferring dependency. Among men, injurious fall was associated with dressing dependency, and among women, impaired hearing was associated with transferring dependency (see Table 4).

## 6. Discussion

To our knowledge, this study is the first to assess the prevalence of geriatric conditions and its relationship to disability among older adults (≥60 years) in a national community-based sample in India in 2017–2018. This study found that the proportion of elderly people having one or more of eight geriatric conditions (55.6%) was higher than in a study among older adults in the USA (49.9%, one or more of seven geriatric conditions) [3]. Compared to the USA study [3], this investigation found a higher prevalence of some geriatric conditions, such as underweight (26.9% vs. 2.9%), impaired cognition (9.3% vs. 7.3%), impaired vision (13.7% vs. 7.3%), and dizziness (14.5% vs. 13.4%), and a lower prevalence in other geriatric conditions, such as injurious falls (5.7% vs. 9.6%), incontinence (4.1% vs. 12.7%), and impaired hearing (9.6% vs. 25.7%). Some of these differences may be related to differences in measurement, such as for hearing impairment. However, the high prevalence of underweight in India seems to show the high undernutrition in India. In a national study among older adults (≥50 years) in India in 2007, the prevalence of underweight (<18.5 kg/m^2^) was 35% [18] and may be attributed to food insecurity [19]. In a study among geriatric medicine out-patients (≥65 years) in Chennai, India, “the prevalence of malnutrition in elderly patients was 31.3%, and those at risk of malnutrition was 54.8%”, emphasizing the need to prioritise medical evaluation and management of malnutrition [20].

Consistent with previous research among older adults in the USA [3], ADL dependencies increased with the higher number of geriatric conditions. ADL dependencies also increased with a higher number of chronic conditions, but to a lesser extent than higher number of geriatric conditions, which also concurs with previous research [3,8]. In particular, the chronic conditions of heart disease, stroke, and bone or joint disorder increased the odds of ADL dependencies. Similar results were found in a study among older adults in the USA [3]. The finding that geriatric conditions contribute to a larger extent than chronic conditions to ADL dependencies, is important for public health and health management in the older population.

The study found that vision impairment and impaired cognition were strongly associated with ADL dependencies, as well as dressing, bathing, eating, transferring, and toileting dependencies. In a systematic review, moderate to strong evidence was found that age, cognitive impairment, and vision impairment are prognostic factors of disability [21,22,23]. Age and cognitive function need to be taken into account in patient care as they significantly increase the risk of disability [22]. Impaired vision may be modifiable to some degree by secondary care (surgery, ophthalmological devices, and improved home lighting) [7,22]. Potentially, multiple behavioural risk factor interventions may assist in the prevention of functional and cognitive decline among older adults in the general population [24]. In addition, hearing impairment was associated with ADL dependencies in this study, which is consistent with some previous studies [21]. Similar to some previous research [3,25,26,27], we did not find associations between the geriatric conditions of injurious falls, dizziness and incontinence and ADL dependencies. However, in gender-stratified analyses, incontinence was among men but not among women associated with dressing, bathing, eating, transferring and toileting dependency.

Study limitations include the cross-sectional design and the assessment of some variables by self-report. A bias may be less for diagnosed chronic conditions than for self-reported health. Some geriatric conditions, such as delirium, pressure sores and frailty, were not included in this study and should be included in future research. Furthermore, the study focused on community-dwelling older adults and excluded institutionalised people.

## 7. Conclusions

More than half of the older adults in India had at least one geriatric condition. The prevalence of geriatric conditions was as high as the prevalence of chronic conditions. A higher number of geriatric conditions, and a higher number of chronic conditions, heart disease, stroke, and bone or joint disorder, were associated with ADL dependencies. Geriatric conditions should be included in health care management.

## Figures and Tables

**Table 1 geriatrics-06-00071-t001:** Sample and geriatric condition characteristics among older adults (≥60 years) in India, 2017–2018.

Variable	Sample	Geriatric Conditions				*p*-Value	Geriatric Conditions		*p*-Value
		0	1	2	≥3		≥2 Men	≥2 Women	
	N (%)	%	%	%	%		%	%	
Sociodemographic factors									
All	31477	44.3	32.7	15.9	7.1		19.8	26.3	<0.001
Age in years						<0.001			0.021
60–69	18979 (58.5)	48.7	32.1	13.9	5.3		15.8	22.5	
70–79	9108 (30.2)	40.1	34	18.1	7.8		22.4	29.8	
80 plus	3390 (11.3)	26.5	33.3	22.5	17.7		36.9	43.8	
Sex						<0.001	---	---	---
Female	15106 (52.5)	41	32.7	17.5	8.8				
Male	16371 (47.5)	47.5	32.8	14.3	5.5				
Marital status						<0.001			<0.001
Not married	11545 (38.4)	37.5	33.6	19.2	9.7		27	29.6	
Married	19926 (61.6)	47.9	32.2	14.1	5.7		18.1	22.7	
Education						<0.001			<0.001
No schooling	16894 (56.5)	33.8	35.2	20.9	10.1		29.7	31.7	
≥1 year	14583 (43.5)	56.5	29.8	10	3.7		13.9	13	
Socioeconomic status						<0.001			0.183
Low	10664 (39.1)	33.4	35.7	20.8	10.1		28.4	33.2	
Medium	11757 (46.7)	47	32.5	14	6.5		18.4	22.7	
High	7836 (24.2)	56.2	29.3	11	3.5		10.5	19.1	
Residential status						<0.001			0.011
Rural	20730 (70.5)	37.6	35.5	18.3	8.7		23.4	30.8	
Urban	10747 (29.5)	60.3	26.2	10	3.4		10.2	16.4	
Religion						0.747			0.656
Hindu	23047 (82.2)	44.5	32.6	15.8	7.2		19.7	26.2	
Muslim	3732 (11.3)	43	33.1	16.4	7.5		19.2	29.2	
Christian	3151 (2.9)	41.3	37.3	16.5	4.9		19.4	23.2	
Sikh	979 (1.9)	46.6	32.5	15.7	5.2		19.9	22	
Other	567 (1.7)	42.5	31.4	17.7	8.4		27.3	24.8	
Health factors									
Heart disease	1572 (5.2)	51.3	27.3	12.3	9.1	0.011	18.9	25.5	0.033
Stroke	842 (2.7)	33.7	35.6	19.7	10.9	<0.001	30.4	31.1	0.002
Angina	2805 (9.2)	37.3	32.4	20.5	9.8	<0.001	26	33.9	0.106
Chronic lung disease	2371 (8.5)	32.2	35.2	22.3	10.3	<0.001	32.1	33.3	<0.001
Bone or joint disorder	5586 (19.7)	43.3	29.9	17.2	9.6	<0.001	26.4	27.1	0.39
Diabetes	4862 (14.3)	58.6	27.5	9.7	4.1	<0.001	12.7	15	0.221
Cancer	238 (0.7)	44.5	29.2	20.4	5.9	0.748	14.8	37	0.118
No of chronic conditions						<0.001			0.323
0	17899 (56.4)	42.9	34.4	16.2	6.6		19	26.6	
1	9666 (30.8)	46.9	31	14.7	7.4		18.7	25.7	
2 plus	3785 (12.8)	44.5	29	17.5	8.9		26.8	26.2	
≥1 ADL dependencies						<0.001			0.235
No	29137 (92.6)	45.3	32.8	15.3	6.6		18.9	25	
Yes	2203 (7.4)	22.9	31.9	26.9	18.3		41.1	48.1	

ADL: Activities of Daily Living.

**Table 2 geriatrics-06-00071-t002:** Geriatric conditions by age group and sex.

Variable	Age in Years			
All	60–69	70–79	80 plus	60+
	%	%	%	%
Injurious falls	5.8	5.1	6.7	5.7
Impaired cognition	6.9	10.7	19.9	9.3
Underweight	23.7	28.7	39.9	26.9
Dizziness	14.1	15.3	14.6	14.5
Incontinence	2.9	5.0	7.9	4.1
Impaired vision	9.9	15.9	27.4	13.7
Impaired hearing	7.1	12.1	16.1	9.6
Major depressive disorder	8.0	7.9	10.3	8.2
Men				
Injurious falls	5.5	4.6	6.3	5.3
Impaired cognition	3.3	5.3	11.7	4.7
Underweight	25.0	30.5	42.0	28.4
Dizziness	11.2	12.8	9.7	11.5
Incontinence	3.1	4.0	7.3	4.4
Impaired vision	8.5	13.1	25.2	11.7
Impaired hearing	6.9	12.7	16.9	9.8
Major depressive disorder	7.3	6.4	9.1	7.2
Women				
Injurious falls	6.0	5.6	7.1	6.0
Impaired cognition	10.3	16.1	28.3	13.6
Underweight	22.6	27.0	38.0	25.4
Dizziness	16.6	17.7	18.8	17.2
Incontinence	3.1	4.0	7.3	3.8
Impaired vision	11.1	18.6	29.3	15.4
Impaired hearing	7.3	11.5	15.5	9.5
Major depressive disorder	8.6	9.3	11.4	9.1

**Table 3 geriatrics-06-00071-t003:** Odds ratios for Activities of Daily Living Dependency.

Variable	Model 1	Model 2	Model 3	Model 4
All	COR (95% CI)	AOR (95% CI) ^a^	AOR (95% CI) ^b^	AOR 95% CI) ^c^
Geriatric conditions				
0	1 (Reference)	1 (Reference)	1 (Reference)	1 (Reference)
1	1.93 (1.49, 2.48) ***	1.90 (1.45, 2.49) ***	1.93 (1.47, 2.54) ***	1.85 (1.41, 2.44) ***
2	3.46 (2.62, 4.57) ***	3.43 (2.54, 5.65) ***	3.35 (2.46, 4.56) ***	3.19 (2.35, 4.34) ***
≥3	5.49 (3.94, 7.85) ***	5.31 (3.65, 7.72) ***	5.00 (3.36, 7.43) ***	4.64 (3.08, 7.00) ***
Chronic conditions				
0			1 (Reference)	
1			1.69 (1.32, 2.16) ***	
≥2			2.75 (2.06, 3.67) ***	
Type of chronic condition				
Heart disease				1.90 (1.31, 2.75) ***
Stroke				2.69 (1.72, 4.20) ***
Angina				1.17 (0.81, 1.68)
Chronic lung disease				1.27 (0.93, 1.75)
Bone or joint disorder				1.88 (1.48, 2.40) ***
Diabetes				1.03 (0.77, 1.38)
Cancer				1.80 (0.88, 3.66)
Men				
Geriatric conditions				
0	1 (Reference)	1 (Reference)	1 (Reference)	1 (Reference)
1	2.44 (1.60, 3.70) ***	2.37 (1.53, 3.66) ***	2.36 (1.53, 3.64) ***	2.31 (1.49, 3.59) ***
2	3.99 (2.61, 6.10) ***	3.95 (2.44, 6.40) ***	3.76 (2.33, 6.06) ***	3.62 (2.23, 5.86) ***
≥3	6.74 (4.03, 11.27) ***	6.80 (3.88, 11.92) ***	5.86 (3.37, 10.18) ***	5.43 (3.08, 9.59) ***
Chronic conditions				
0			1 (Reference)	
1			1.88 (1.29, 2.73) ***	
≥2			3.24 (2.20, 4.78) ***	
Type of chronic condition				
Heart disease				2.14 (1.24, 3.68) **
Stroke				2.76 (1.52, 5.02) ***
Angina				1.34 (0.71, 2.52)
Chronic lung disease				1.17 (0.75, 1.81)
Bone or joint disorder				1.70 (1.18, 2.44) **
Diabetes				1.25 (0.84, 1.85)
Cancer				1.17 (0.39, 3.50)
Women				
Geriatric conditions				
0	1 (Reference)	1 (Reference)	1 (Reference)	1 (Reference)
1	1.56 (1.13, 2.15) **	1.58 (1.13, 2.22) **	1.61 (1.14, 2.28) **	1.55 (1.11, 2.18) *
2	2.97 (2.08, 4.25) ***	3.03 (2.08, 4.42) ***	3.00 (2.05, 4.39) ***	2.86 (1.97, 4.16) ***
≥3	4.46 (2.69, 7.40) ***	4.39 (2.57, 7.51) ***	4.30 (2.47, 7.50) ***	4.06 (2.30, 7.16) ***
Chronic conditions				
0			1 (Reference)	
1			1.54 (1.11, 2.15) **	
≥2			2.34 (1.58, 3.45) ***	
Type of chronic condition				
Heart disease				1.72 (1.05, 2.83) *
Stroke				1.75 (1.46, 5.16) **
Angina				1.04 (0.68, 1.59)
Chronic lung disease				1.41 (0.92, 2.17)
Bone or joint disorder				1.98 (1.46, 2.67) ***
Diabetes				0.87 (0.58, 1.31)
Cancer				2.25 (0.89, 5.72)

COR: Crude odds ratio, AOR: Adjusted odds ratio; CI = Confidence interval; *** *p* < 0.001; ** *p* < 0.01; * *p* < 0.05; ^a^ Adjusted for sociodemographic factors, ^b^ Adjusted for sociodemographic factors and chronic conditions, ^c^ Adjusted for sociodemographic factors and type of chronic conditions.

**Table 4 geriatrics-06-00071-t004:** Associations between geriatric conditions and Activities of Daily Living Dependency.

**All**	**Prevalence**	**Model 1: 1 ADL Dependencies ^a^**	**Prevalence**	**Model 2: Dressing ^a^**	**Prevalence**	**Model 3: Bathing ^a^**
	%	AOR (95% CI)	%	AOR (95% CI)	%	AOR (95% CI)
Injurious falls	8.6	1.40 (0.99, 1.97)	9.8	1.33 (0.72, 2.44)	9.5	1.11 (0.60, 2.03)
Impaired cognition	19.1	2.04 (1.52, 2.76) ***	28.5	4.19 (2.59, 6.77) ***	27.2	3.61 (2.27, 5.76) ***
Underweight	32.4	1.03 (0.81, 1.30)	35.9	1.04 (0.68, 1.59)	38.5	1.06 (0.71, 1.57)
Dizziness	21.4	1.26 (0.95, 1.67)	22.8	1.05 (0.63, 1.75)	23.6	1.34 (0.86, 2.09)
Incontinence	9.1	1.81 (0.99, 1.97)	11.6	2.16 (1.21, 3.85) **	12.0	2.83 (1.67, 4.78) ***
Impaired vision	30.9	2.68 (2.10, 3.41) ***	37.4	3.54 (2.31, 5.43) ***	36.9	3.58 (2.40, 5.34) ***
Impaired hearing	16.6	1.75 (1.32, 2.32) ***	16.3	0.99 (0.62, 1.59)	15.7	0.99 (0.64, 1.52)
Major depressive disorder	19.0	1.90 (1.31, 2.77) ***	23.6	2.89 (1.59, 5.25) ***	22.9	2.18 (1.18, 4.05) *
**All**	**Prevalence**	**Model 4: Eating ^a^**	**Prevalence**	**Model 5: Transferring ^a^**	**Prevalence**	**Model 6: Toileting ^a^**
	%	AOR (95% CI)	%	AOR (95% CI)	%	AOR (95% CI)
Injurious falls	10.1	1.15 (0.57, 2.32)	9.0	1.12 (0.69, 1.80)	8.7	1.32 (0.88, 1.98)
Impaired cognition	25.6	2.61 (1.67, 4.06) ***	20.2	2.23 (1.42, 3.50) ***	20.3	2.07 (1.46, 2.95) ***
Underweight	36.1	0.96 (0.62, 1.49)	32.6	1.11 (0.77, 1.59)	33.0	1.04 (0.78, 1.39)
Dizziness	20.8	0.93 (0.58, 1.47)	22.1	1.26 (0.82, 1.93)	21.6	1.03 (0.73, 1.45)
Incontinence	9.6	1.85 (1.04, 3.27) *	9.4	1.76 (1.07, 2.90) *	9.6	1.71 (1.08, 2.69) *
Impaired vision	37.7	4.91 (3.35, 7.19) ***	32.2	3.01 (2.07, 4.38) ***	33.3	2.97 (2.22, 3.96) ***
Impaired hearing	16.0	1.23 (0.79, 1.92)	16.9	1.84 (1.22, 2.77) **	17.2	1.74 (1.28, 2.38) ***
Major depressive disorder	24.1	2.58 (1.37, 4.86) **	21.5	2.19 (1.25, 3.84) **	21.1	1.97 (1.26, 3.08) **
**Men**	**Prevalence**	**Model 1: 1 ADL dependencies ^a^**	**Prevalence**	**Model 2: Dressing ^a^**	**Prevalence**	**Model 3: Bathing ^a^**
	%	AOR (95% CI)	%	AOR (95% CI)	%	AOR (95% CI)
Injurious falls	9.3	1.58 (0.93, 2.70)	12.1	2.48 (1.15, 5.36) *	11.0	1.60 (0.78, 3.28)
Impaired cognition	10.6	2.42 (1.38, 4.23) **	18.3	8.64 (4.28, 17.46) ***	17.6	6.56 (3.45, 12.51) ***
Underweight	31.2	0.91 (0.63, 1.32)	33.4	0.74 (0.40, 1.35)	35.4	0.77 (0.46, 1.30)
Dizziness	19.6	1.16 (0.74, 1.82)	21.1	0.73 (0.35, 1.51)	22.2	1.36 (0.73, 2.53)
Incontinence	11.2	2.49 (1.45, 4.29) ***	14.2	3.55 (1.66, 7.57) ***	13.7	3.83 (1.89, 7.76) ***
Impaired vision	28.1	3.09 (2.05, 4.68) ***	33.9	4.92 (2.34, 10.35) ***	32.4	4.48 (2.36, 8.50) ***
Impaired hearing	16.8	1.75 (1.16, 2.64) **	17.2	1.02 (0.48, 2.20)	15.4	1.09 (0.61, 1.97)
Major depressive disorder	19.0	2.18 (1.32, 3.59) **	23.2	2.71 (1.16, 6.33) *	22.0	1.63 (0.79, 3.39)
**Men**	**Prevalence**	**Model 4: Eating ^a^**	**Prevalence**	**Model 5: Transferring ^a^**	**Prevalence**	**Model 6: Toileting ^a^**
	%	AOR (95% CI)	%	AOR (95% CI)	%	AOR (95% CI)
Injurious falls	11.2	1.30 (0.52, 3.21)	10.6	1.81 (0.89, 3.68)	10.0	1.46 (0.79, 2.69)
Impaired cognition	17.2	5.64 (2.85, 11.16) ***	11.9	3.41 (1.63, 7.14) ***	11.5	2.27 (1.18, 4.35) *
Underweight	31.1	0.75 (0.41, 1.40)	29.4	0.79 (0.45, 1.40)	31.6	0.78 (0.50, 1.23)
Dizziness	21.4	0.99 (0.51, 1.94)	22.8	1.44 (0.79, 2.65)	21.1	1.14 (0.69, 1.85)
Incontinence	11.9	2.26 (1.10, 4.64) *	11.2	2.20 (1.11, 4.39) *	12.6	2.44 (1.33, 4.49) **
Impaired vision	35.6	7.08 (3.83, 13.09) ***	31.2	4.61 (2.56, 8.29) ***	30.0	3.79 (2.36, 6.05) ***
Impaired hearing	15.0	0.86 (0.43, 1.75)	14.9	1.38 (0.72, 2.64)	17.6	1.92 (1.23, 2.97) **
Major depressive disorder	25.3	2.66 (1.16, 6.13) *	20.8	1.81 (0.89, 3.68)	21.5	1.88 (1.06, 3.34) *
**Women**	**Prevalence**	**Model 1: 1 ADL dependencies ^a^**	**Prevalence**	**Model 2: Dressing ^a^**	**Prevalence**	**Model 3: Bathing ^a^**
	%	AOR (95% CI)	%	AOR (95% CI)	%	AOR (95% CI)
Injurious falls	8.1	1.26 (0.94, 3.17)	8.0	0.78 (0.26, 2.31)	8.2	0.74 (0.25, 2.20)
Impaired cognition	25.3	1.90 (1.31, 2.75) ***	35.8	3.01 (1.58, 5.71) ***	35.2	2.64 (1.35, 5.17) **
Underweight	33.3	1.15 (0.79, 1.67)	37.4	1.32 (0.68, 2.54)	40.8	1.34 (0.68, 2.63)
Dizziness	22.6	1.28 (0.87, 1.88)	24.0	1.20 (0.60, 2.39)	24.7	1.32 (0.67, 2.59)
Incontinence	7.6	1.24 (0.72, 2.12)	9.7	1.18 (0.52, 2.67)	10.7	1.75 (0.81, 3.81)
Impaired vision	32.7	2.39 (1.66, 3.43) ***	39.9	2.90 (1.58, 5.33) ***	40.4	3.03 (1.63, 5.65) ***
Impaired hearing	16.5	1.75 (1.18, 2.61) ***	15.6	0.95 (0.51, 1.79)	16.0	0.84 (0.44, 1.61)
Major depressive disorder	19.0	1.73 (0.94, 3.17)	23.8	2.85 (1.17, 6.99) *	23.6	2.59 (0.99, 6.78)
**Women**	**Prevalence**	**Model 4: Eating ^a^**	**Prevalence**	**Model 5: Transferring ^a^**	**Prevalence**	**Model 6: Toileting ^a^**
	%	AOR (95% CI)	%	AOR (95% CI)	%	AOR (95% CI)
Injurious falls	9.2	1.05 (0.36, 3.04)	7.8	0.95 (0.48, 1.89)	7.7	1.20 (0.68, 2.15)
Impaired cognition	32.5	1.78 (0.98, 3.25)	26.6	1.84 (1.01, 3.34) *	26.6	1.99 (1.27, 3.11) **
Underweight	39.5	1.16 (0.56, 2.42)	34.9	1.44 (0.82, 2.53)	34.0	1.31 (0.85, 2.02)
Dizziness	20.3	0.88 (0.44, 1.74)	21.7	1.12 (0.60, 2.09)	22.0	0.93 (0.57, 1.53)
Incontinence	7.9	1.05 (0.36, 3.04)	8.1	1.26 (0.64, 2.51)	7.4	1.03 (0.54, 1.96)
Impaired vision	39.2	3.69 (2.03, 6.69) ***	32.9	2.13 (1.14, 3.96) *	35.5	2.45 (1.58, 3.80) ***
Impaired hearing	16.8	1.61 (0.89, 2.89)	18.3	2.29 (1.31, 3.98) **	16.9	1.58 (0.99, 2.53)
Major depressive disorder	23.2	2.53 (0.92, 6.93)	22.0	2.46 (1.05, 5.81) *	20.8	1.99 (0.99, 4.02)

^a^ Each model was adjusted for sociodemographic factors and geriatric conditions; AOR: Adjusted odds ratio; CI = Confidence interval; *** *p* < 0.001; ** *p* < 0.01; * *p* < 0.05.

## Data Availability

The data are available at the The Gateway to Global Aging Data (www.g2aging.org accessed on 20 July 2021).

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
