# Peer review of "Geriatric Conditions and Functional Disability among a National Community-Dwelling Sample of Older Adults in India in 2017–2018"

_geriatrics, 2021, doi:10.3390/geriatrics6030071_

Round 1

Reviewer 1 Report

The paper is pretty good. Thank you for the work done and for the chosen topic. It is very interesting as well as necessary.

Some changes and suggestions for improvement are presented. Most of them correspond to typographical errors and suggestions for style and presentation of information. I hope these will be useful.

The corrections are organized by sections and identifying the specific line in which they are found to make it easier for the author to identify them.

ABSTRACT

Line 13: it should be “prevalence of” instead of “prevalence”.

Line 21: the results should be ordered from highest to lowest. However, the ADL dependencies percentage is at the end. I suggest to order it as mentioned.

Line 24: there should be a . before "in a model adjusted...." in order to separate that long sentence into two shorter ones. This will make it easier for the reader to read that section.

INTRODUCTION

Line 41: the citation of the World Health Organization in this line is unnecessary. I would remove this first citation so that it is not redundant.

Line 43: I would remove the . before ", simply mark the beginning and end of the quote with " ".

Line 56: it would be simpler to order these data from highest to lowest, unless the order followed is for some other specific reason.

METHOD

Line 84: explain better what the Conbrach alpha data refers to. It is not clear what it contributes to the presentation of the variable outcome.

RESULTS

Line 131: again, it should be better if ordered from highest to lowest.

Line 160: a . is needed to make the sentence clearer. Perhaps after "ADL dependencies" and before "in a model adjusted..." would be a good place.

Line 173: there is only information on impaired cognition and impaired vision, but what about the rest of the geriatric conditions? More information is needed to explain Table 4. 

Line 175: in Table 4, it should be “prevalence” instead of “preva-lence”. Is must be a typo. The second part of the Table, when models 4 to 6 are presented, should be in bold.

DISCUSSION

Line 179: information on what the prevalence refers to is missing.

Line 183: the second citation from the USA study is not necessary. It is understood that it is referring to the study cited immediately before.

Line 207: at the beginning of the paragraph, there are a couple of direct quotation from other authors but there is no novel contribution. It would be interesting to include some reflection on what these findings in the literature contribute to this study.

Line 233: the " at the end of the acknowledgements section should be deleted.

Author Response

Reviewer I
The paper is pretty good. Thank you for the work done and for the chosen topic. It is very interesting as well as necessary.
Some changes and suggestions for improvement are presented. Most of them correspond to typographical errors and suggestions for style and presentation of information. I hope these will be useful.
The corrections are organized by sections and identifying the specific line in which they are found to make it easier for the author to identify them.
ABSTRACT
Line 13: it should be “prevalence of” instead of “prevalence”.
Response: Corrected
Line 21: the results should be ordered from highest to lowest. However, the ADL dependencies percentage is at the end. I suggest to order it as mentioned.
Response: Corrected
Line 24: there should be a . before "in a model adjusted...." in order to separate that long sentence into two shorter ones. This will make it easier for the reader to read that section.
Response: Corrected
INTRODUCTION
Line 41: the citation of the World Health Organization in this line is unnecessary. I would remove this first citation so that it is not redundant.
Response: Corrected
Line 43: I would remove the . before ", simply mark the beginning and end of the quote with " ".
Response: Corrected
Line 56: it would be simpler to order these data from highest to lowest, unless the order followed is for some other specific reason.
Response: Corrected

METHOD
Line 84: explain better what the Conbrach alpha data refers to. It is not clear what it contributes to the presentation of the variable outcome.
Response: added as below
Cronbach alpha was 0.86 for the ADL scale in this study
RESULTS
Line 131: again, it should be better if ordered from highest to lowest.
Response: corrected
Line 160: a . is needed to make the sentence clearer. Perhaps after "ADL dependencies" and before "in a model adjusted..." would be a good place.
Response: corrected
Line 173: there is only information on impaired cognition and impaired vision, but what about the rest of the geriatric conditions? More information is needed to explain Table 4. 
Response: more is added, as below
The odds of ADL dependencies increased with impaired cognition, impaired vision, impaired hearing, and major depressive disorder in the overall and male sex model, while in the female sex model ADL dependencies increased with impaired cognition, impaired vision, and impaired hearing. Impaired cognition, incontinence, impaired vision and major depressive disorder were positively associated with dressing, bathing, eating, transferring, and toileting dependency. In addition, impaired hearing was associated with transferring and toileting dependency. 
In gender stratified analysis, incontinence and major depressive disorder were among men but not among women associated with ADL dependencies. Incontinence was among men but not among women associated with dressing, bathing, eating, transferring and toileting dependency. Major depressive disorder was among both men and women associated with dressing dependency, among men but not women with eating and toileting dependency and among women but not men with transferring dependency. Among men, injurious fall was associated with dressing dependency, and among women, impaired hearing was associated with transferring dependency
Line 175: in Table 4, it should be “prevalence” instead of “preva-lence”. Is must be a typo. The second part of the Table, when models 4 to 6 are presented, should be in bold.
Response: corrected
DISCUSSION
Line 179: information on what the prevalence refers to is missing.
Response: added
Line 183: the second citation from the USA study is not necessary. It is understood that it is referring to the study cited immediately before.
Response: Removed
Line 207: at the beginning of the paragraph, there are a couple of direct quotation from other authors but there is no novel contribution. It would be interesting to include some reflection on what these findings in the literature contribute to this study.
Response: Added
Line 233: the " at the end of the acknowledgements section should be deleted.
Response: Removed

Reviewer 2 Report

The authors use an extensive database, the Longitudinal Ageing Study in India (LASI) Wave 1, 2017-2018 to address the prevalence of geriatric conditions in India, projected the most populated country in the next coming decades. They relate the 6 items assessed with a disability as measured by DLA, daily life activity.
Despite the big sample size and balance of gender that should allow a gender analysis to dissect different profiles in old women and men, this analysis is lacking and most of the 'noveltty' and scientific soundness is lost. The work could reach that level if the analysis is done. Nowadays it is also hardly acceptable not to do so, when the data is available. Therefore, I can only provide a positive evaluation if the effort is done.

In the discussion, they compare with data from USA. As expected, DLA depended on the number of geriatric conditions more than chronic conditions. The authors should emphasize this fact as it is important for public health and health management in the older population. Also, results also clearly point to malnutrition in older people as an important problem.

In the discussion, there is an abundance of 'quotes' that should better be replaced by integrated sentences. Limitations are identified and depicted.

Specific comments :

Abstract

Lines 13-32 Subparts are no needed to be indicated (you can omit objectives, methods, discussion, etc).

Multidimensional geriatric assessment should also include psychiatric/affective aspects such as mood, anxiety, depression. As depicted in line 55, in India  '1 in 4 suffer from depression'. In fact, in the tables, " Neurological or psychiatric disorder' is listed among the ítems. Please, clarify.

The age for 'aged' population is established at 65 for developed countries and 60 for LMIC. In the text, line 49, the reference for 'age of 60' is clearly stated by reference 5. Also, in the methods, it is indicated that the database included people +45. Please, justify why the age cut of 65 was used in the present work, as this choice results in different prevalence/incidence results.

Methods

Several validated scales are used to assess ADL. Please, indicate which one was used or modified for the present work and depict the scale of quantification

Results

Lack of gender analysis, which is a must.

Discussion

The abundance of quotes. For instance, for a whole sentence that it is a quote: line 207.209 " “Although age and cognitive functioning are not modifiable 207 prognostic factors, they 208 must be taken into account in targeting care as they indicate high-risk for increasing dis209 ability” [16]. Could be replaced by authors' analysis: " A systematic review on prognostic factors of disability in older people provided evidence that…."

Author Response

Reviewer 2
The authors use an extensive database, the Longitudinal Ageing Study in India (LASI) Wave 1, 2017-2018 to address the prevalence of geriatric conditions in India, projected the most populated country in the next coming decades. They relate the 6 items assessed with a disability as measured by DLA, daily life activity.
Despite the big sample size and balance of gender that should allow a gender analysis to dissect different profiles in old women and men, this analysis is lacking and most of the 'noveltty' and scientific soundness is lost. The work could reach that level if the analysis is done. Nowadays it is also hardly acceptable not to do so, when the data is available. Therefore, I can only provide a positive evaluation if the effort is done.
Response: Gender stratified analyses are added

In the discussion, they compare with data from USA. As expected, DLA depended on the number of geriatric conditions more than chronic conditions. The authors should emphasize this fact as it is important for public health and health management in the older population. Also, results also clearly point to malnutrition in older people as an important problem.

Response: this added

In the discussion, there is an abundance of 'quotes' that should better be replaced by integrated sentences. Limitations are identified and depicted.
Response: This is corrected
Specific comments :
Abstract
Lines 13-32 Subparts are no needed to be indicated (you can omit objectives, methods, discussion, etc).
 Response: removed
Multidimensional geriatric assessment should also include psychiatric/affective aspects such as mood, anxiety, depression. As depicted in line 55, in India  '1 in 4 suffer from depression'. In fact, in the tables, " Neurological or psychiatric disorder' is listed among the ítems. Please, clarify.
 Response: We have now included major depressive disorder under geriatric conditions
The age for 'aged' population is established at 65 for developed countries and 60 for LMIC. In the text, line 49, the reference for 'age of 60' is clearly stated by reference 5. Also, in the methods, it is indicated that the database included people +45. Please, justify why the age cut of 65 was used in the present work, as this choice results in different prevalence/incidence results.
Response: This is changed to 60 years and above

Methods
Several validated scales are used to assess ADL. Please, indicate which one was used or modified for the present work and depict the scale of quantification
Response: below is added
 Activities of Daily Living (ADL) (6 items) included difficulties with dressing, ambulating, bathing, eating, transferring and toileting (Yes/No) [10]; Cronbach alpha was 0.86 for the ADL scale in this study. The ADL instrument has shown acceptable validity for the geriatric population in India (16).
Results
Lack of gender analysis, which is a must.
Response: All tables have now included a gender stratified analysis

Discussion
The abundance of quotes. For instance, for a whole sentence that it is a quote: line 207.209 " “Although age and cognitive functioning are not modifiable 207 prognostic factors, they 208 must be taken into account in targeting care as they indicate high-risk for increasing dis209 ability” [16]. Could be replaced by authors' analysis: " A systematic review on prognostic factors of disability in older people provided evidence that…."

Response: This is reworked

Round 2

Reviewer 2 Report

The authors have properly addressed all the issues raised, the most important one on psychological/affective aspects of multidimensional geriatric analysis and the gender issue that was a must,
The clinical interest of the present rewritten version has increased and  provides clear women/men differences in the comorbidity patterns that will contribute to better use of the new knowledge.
It's a pleasure to endorse their work.